# Synergism Potentiates Oxidative Antiproliferative Effects of Naringenin and Quercetin in MCF-7 Breast Cancer Cells

**DOI:** 10.3390/nu14163437

**Published:** 2022-08-21

**Authors:** Mahasin Abdel Rhman, Nikita Devnarain, Rene Khan, Peter M. O. Owira

**Affiliations:** Molecular and Clinical Pharmacology Research Laboratory, Department of Pharmacology, Discipline of Pharmaceutical Sciences, University of Kwazulu-Natal, P.O. Box X5401, Durban 4000, South Africa

**Keywords:** quercetin, naringenin, breast cancer, oxidative stress, apoptosis

## Abstract

Breast cancer (BC) is the most frequently diagnosed type of cancer as of 2020. Quercetin (Que) and Naringenin (Nar) are predominantly found in citrus fruits and vegetables and have shown promising antiproliferative effects in multiple studies. It is also known that the bioactive effects of these flavonoids are more pronounced in whole fruit than in isolation. This study investigates the potential synergistic effects of Que and Nar (CoQN) in MCF-7 BC cells. MCF-7 cells were treated with a range of concentrations of Que, Nar or CoQN to determine cell viability. The IC_50_ of CoQN was then used to investigate caspase 3/7 activity, Bcl-2 gene expression, lipid peroxidation and mitochondrial membrane potential to evaluate oxidative stress and apoptosis. CoQN treatment produced significant cytotoxicity, reduced Bcl-2 gene expression and increased caspase 3/7 activity compared to either Nar or Que. Furthermore, CoQN significantly increased lipid peroxidation and reduced mitochondrial membrane potential (MMP) compared to either Nar or Que. Therefore, CoQN treatment has potential pharmacological application in BC chemotherapy by inducing oxidative stress and apoptosis in MCF-7 BC cells. The results of this study support the increased consumption of whole fruits and vegetables to reduce cell proliferation in cancer.

## 1. Introduction

By 2020, breast cancer (BC) was the most frequently diagnosed type of cancer and the fifth leading cause of cancer-related deaths globally in women [1]. Despite technological advancements in the diagnosis and treatment, BC continues to burden the healthcare systems due to increased prevalence associated with socioeconomic disadvantages and continued exposure to risk factors [2]. BC mainly originates in the lobules or ducts of the breast and can be non-invasive or invasive, spreading to the lymph nodes and metastasizing to other parts of the body [3]. Its treatment requires a multidisciplinary approach involving surgery, radiation, chemotherapy, endocrine therapy and human epidermal growth factor receptor 2 (HER2)-targeted therapy depending on staging and molecular subtype [4]. Despite advancements in therapeutic approaches to BC, drug side effects due to narrow therapeutic indices and radiation burden reduce compliance. Therefore, novel therapeutic agents targeting BC cells with less cytotoxicity are required.

Polyphenolic flavonoids are plant-derived bioactive chemical substances [5] with anticancer, antioxidant, anti-inflammatory, antidiabetic and cardioprotective properties [6]. Flavonoids have been shown to exert their chemotherapeutic effects in various cancers by multiple pathways, including increased oxidative stress and apoptosis leading to reduced cell proliferation, invasion and migration [7]. Quercetin (Que) and Naringenin (Nar) are significant flavonoids found in many plant sources, including onions, broccoli, berries and citrus fruits [8,9]. Nar is the aglycone form of the flavanone glycoside, naringin [9]. The pharmacological effects of Que or Nar have previously been investigated in vitro in MCF-7 and MDA-MB-231 cell lines and in vivo using xenografted or genetically modified BC mice models with promising results [10,11,12]. However, the administration of flavonoids usually requires high concentrations to produce a cytotoxic effect in cancer cells due to their low efficacy [13]. Co-administered flavonoids could have synergistic effects at lower concentrations and enhance selective toxicity in cancer cells [14,15,16]. We posit that plant bioactive chemical ingredients are more effective in their natural composition than when isolated. Compounds which are constituents of a particular plant can influence biological processes in a synergistic manner.

The intrinsic pathway of apoptosis (programmed cell death) involves non-receptor mediated intracellular signaling generated upon stimuli. In contrast, the extrinsic pathway requires binding an extracellular ligand to death receptors anchored in the cellular membrane [17]. Anti-apoptotic protein, Bcl-2, plays an essential role in preventing intrinsic pathway-induced apoptosis by inhibiting caspase cascade activation [18]. Bcl-2 protein expression is usually upregulated in cancerous cells resulting in increased proliferation and reduced apoptosis [19]. Flavonoids have been reported to exert their antiproliferative effects by downregulation of Bcl-2. This leads to the activation of the proapoptotic proteins of the Bcl-2 family, Bax and Bak, allowing their aggregation in the mitochondrial membrane. Subsequently, cytochrome complex (cyt-c) is released from the mitochondrion into the cytoplasm, which activates the apoptotic peptidase activating factor (Apaf1) protein. This activates caspase 9, which initiates the caspase cascade via the intrinsic pathway, leading to the induction of apoptosis [20] (Figure 1).

Reactive oxygen species (ROS) are harmful free radicals, including hydroxyl radicals (•OH) and superoxide radicals (O_2_^•−^), or aid in the formation of free radicals, such as hydrogen peroxide (H_2_O_2_) and singlet oxygen (^1^O_2_). They are counteracted and neutralized by enzymatic or non-enzymatic antioxidants [21]. An imbalance between ROS production and antioxidant activity results in oxidative stress and damage to cellular components, such as lipid membranes and nuclear and mitochondrial DNA [22]. Flavonoids have been shown to exert pro-oxidant effects in various types of cancer cells by several mechanisms, including auto-oxidation generating free radicals, the depletion of antioxidant stores and the inhibition of enzymatic antioxidant activity [23]. Oxidative stress can also cause a reduction in mitochondrial membrane potential (MMP) due to mitochondrial membrane damage. This leads to mitochondrial dysfunction, reduced Bcl-2 activity and the release of cyt-c into the cytoplasm, which activates the caspase cascade through the intrinsic apoptotic pathway [24,25] (Figure 1). Therefore, increased lipid peroxidation and reduced MMP are valuable indicators of oxidative stress-mediated apoptosis.

Therefore, this study aimed to examine the synergistic effect of Que and Nar co-administration (CoQN) compared to either flavonoid alone in MCF-7 BC cell lines. Cell viability, lipid peroxidation, mitochondrial dysfunction and apoptosis of MCF-7 BC cells were used as efficacy indices.

## 2. Materials and Methods

### 2.1. Cell Culture

MCF-7 cell lines were purchased from Cellonex^®^ (Johannesburg). Cell lines were grown to 80% confluency in Dulbecco’s modified Eagles’ medium (DMEM) supplemented with 10% (*v/v*) fetal bovine serum, 1% (*v/v*) penicillin-streptomycin antibiotics, 1% (*v/v*) L-glutamine and 1% (*v/v*) HEPES buffer in 25 cm^2^ flasks. The cell cultures were maintained at 37 °C in a humidified 5% CO_2_ incubator. The media was changed every 24 to 48 h. All the work was carried out under aseptic techniques and cell culture protocol.

### 2.2. Cell Treatments

Que was purchased from Sigma-Aldrich Pty Ltd. (Johannesburg, South Africa; Pcode: 1002181950; CAS number: 117-39-5; Lot ID: SLBM7336V). Nar was purchased from Sigma-Aldrich Pty Ltd. (Johannesburg, South Africa; Pcode: 1002456338; CAS number: 67604-48-2; Lot ID: MKBW8640V). Dimethyl sulfoxide (DMSO) was used to prepare 10 mg/mL stock concentrations of Que and Nar. The cells were then exposed to either Que or Nar only or combined at concentrations required for assays with a final DMSO concentration of less than 0.01%.

### 2.3. Cell Viability Assays

Methylthiazol tetrazolium bromide (MTT) assay was used to determine MCF-7 cell viability. MCF-7 cells (1.5 × 10^4^/well) were seeded in a 96-well plate and incubated for 24 h. Cells were then treated with serial (0–250 µg/mL) concentrations of Que or Nar or a combination at a concentration range of 0–50 µg/mL. Three ratios (50:50, 40:60, and 60:40 for Nar:Que) were used in the combination treatments to assess the most effective ratio. After 24 h of treatment, 20 µL of 5 mg/mL MTT solution and 100 µL of cell culture media were added to the cells and incubated for 4 h. Afterwards, the MTT solution was aspirated and 100 µL of DMSO was added to solubilize the formazan dye. A microplate reader was used to measure absorbance at 570 nm, and cell viability was calculated as a percentage compared to the control.

### 2.4. Apoptosis Detection

Caspase 3/7 fluorometric assay (Biocom^®^, Johannesburg, South Africa) was used to determine apoptosis induction. MCF-7 cells were seeded in a 96-well plate (1.0 × 10^4^/well) and incubated for 24 h. The cells were then treated with Que, Nar or both at a concentration of 44.31 µg/mL for 24 h. The cells were then treated with 100 µL of caspase loading solution for caspase 3/7 detection and incubated for 1 h at room temperature. After that, fluorescence was measured using a microplate reader at an excitation/emission wavelength of 535/620 nm. Results were expressed as relative fluorescence units (RFU).

### 2.5. Real-Time qPCR Analysis

Real-Time qPCR was used to assess the expression of anti-apoptotic Bcl-2 mRNA. Total RNA was extracted from MCF-7 cells previously treated with Nar, Que or both at a concentration of 44.31 µg/mL, respectively, using Quick-RNA Miniprep Kit (Zymo Research Ltd., Johannesburg, South Africa) according to the manufacturer’s instructions. Complementary DNA (cDNA) was synthesized from extracted RNA using the iScript cDNA synthesis kit (BioRad, Hercules, CA, USA). Real-Time qPCR was carried out for Bcl-2 with β-actin as an internal control using the following primers (Biotec^®^, Johannesburg, South Africa): Bcl-2 (F: 5′-GGTGGTGGAGGAACTCTTCA-3′; R: 5′-ATGCCGGTTCAGGTACTCAG-3′), β-actin (F: 5′-GGAGATTACTGCCCTGGCTCCTA-3′; R: 5′-GACTCATCGTACTCCTGCTTGCTG-3′). 

RT-qPCR amplification and measurement were performed using SSo advanced SYBR Green Supermix (BioRad^®^, Hercules, CA, USA) on BioRad qPCR thermocycler in the following conditions: 30 s at 95 °C for polymerase activation followed by 40 cycles of 5 s at 95 °C for denaturation and 15 s at 55 °C for annealing and extension. Gene expression was then analyzed and normalized against the internal control. Results were displayed relative fold change compared to the control (2^−ΔΔCt^) [26].

### 2.6. Thiobarbituric Acid Reactive Substances Assay

Thiobarbituric acid reactive substances (TBARS) assay was used to determine lipid peroxidation. MCF-7 cells were treated with Nar, Que or both at a concentration of 44.31 µg/mL. After that, 2% H_3_PO_4_ was added to cell suspension (2 × 10^5^ cells) in respective treatment tubes, whereas 0.1 M malondialdehyde (MDA) was used as a positive control. After that, 7% H_3_PO_4_ was then added, followed by 400 µL of 0.1 mM thiobarbituric acid (TBA)/Butylated hydroxytoluene. Subsequently, 1 M HCl was added, and the tubes were heated in a water bath for 15 min. Butanol (1.5 mL) was then added, after which the absorbance of the resultant supernatant was measured in a 96-well using a microplate reader at 532 nm and 600 nm. MDA concentrations were then calculated using the extinction coefficient of the MDA-TBA adduct (155 mM^−1^ cm^−1^). 

### 2.7. Mitochondrial Depolarization Detection

MMP was determined using JC-10 fluorometric assay kit (Sigma-Aldrich Ltd., St. Louis, MO, USA) according to the manufacturer’s instructions. MCF-7 cells were seeded in a 96-well plate (1.5 × 10 ^4^/well) in triplicates following treatment with Que, Nar or CoQN at 44.31 µg/mL, respectively, for 24 h. After that, 50 µL of JC-10 loading dye was added to each well and incubated for 30 min in the dark. Red and green fluorescence were then measured using a microplate reader at an excitation/emission wavelength of 540/590 and 490/525, respectively. The ratio of red to green fluorescence was calculated to determine the MMP. Results were expressed as a percentage of control of the red/green fluorescence ratio. 

### 2.8. Statistical Analysis

Statistical analyses were carried out using GraphPad Prism version 9.4.1., GraphPad Software, San Diego, CA, USA. The concentration–response inhibition equation using non-linear regression was used to analyze the IC_50_ in the MTT assay. Statistical significance was assessed by unpaired t-test with Welch’s correction for 2 variable comparisons of each treatment (Que, Nar and CoQN) against the control in the results obtained from the caspase 3/7 assay, Real-Time qPCR, TBARS assay and mitochondrial depolarization assay. Treatment samples (Que, Nar and CoQN) were used in triplicate and experiments were repeated three times for all assays performed. A *p* value of <0.05 was considered to indicate statistical significance.

## 3. Results

### 3.1. MCF-7 Cell Viability

The MTT assay was used to assess the toxicity of Nar, Que or CoQN to MCF-7 BC cells and to determine cellular metabolic activity indicated by mitochondrial dehydrogenase activity. The administration of Que or Nar at a concentration range of 0–250 µg/mL resulted in decreased toxicity to MCF-7 cells compared to CoQN-treated cells. Neither Nar nor Que showed a dose-dependent response, with extrapolated IC_50_ values of 468 µg/mL and 91.10 µg/mL, respectively (Figure 2A). At the 60:40 (N:Q) combination ratio, the calculated IC_50_ (44.31 µg/mL) and dose-dependence were significantly superior to either 50:50 or 40:60 ratios, respectively (Figure 2B). Hence, the 60:40 combination ratio was used in subsequent experiments. 

### 3.2. Apoptosis Induction

Caspase 3/7 fluorometric assay was used to compare treatment effects with 44.31 µg/mL of Que, Nar or both (CoQN) on the apoptosis of MCF-7 BC cells. CoQN significantly (*p* = 0.0052) increased executioner caspase 3/7 activity compared to control or either Nar or Que, signifying increased cell apoptosis (Figure 3). 

The gene expression of Bcl-2 in MCF-7 cells was determined by qPCR following the RNA isolation and cDNA synthesis of all treatment groups to measure the apoptosis pathway. CoQN significantly (*p* < 0.05) reduced the mRNA expression of the anti-apoptotic Bcl-2 gene compared to either control or Que alone, signifying the activation of the intrinsic apoptotic pathway (Figure 4). Bcl-2 gene expression in Nar-treated cells was similar to COQN.

### 3.3. Lipid Peroxidation 

The TBARS assay was used to assess ROS production by quantifying the end-product of lipid peroxidation, MDA, in MCF-7 cells following 24-h treatment with 44.31 µg/mL of Que, Nar or both (CoQN) relative to control. The CoQN treatment significantly (*p* = 0.024) increased MDA concentrations compared to the control, Que or Nar, respectively (Figure 5). Treatments with Que or Nar did not show a significant increase in MDA concentration compared to the control (Figure 5).

### 3.4. Mitochondrial Depolarization

Mitochondrial depolarization in MCF-7 cells was determined by assessing the MMP using the JC-10 dye following treatment with 44.31 µg/mL of Que, Nar or CoQN, respectively. Treatment with CoQN or Nar significantly (*p* < 0.05) decreased MMP compared to controls and Que (Figure 6), suggesting mitochondrial depolarization leading to induction of apoptosis. 

## 4. Discussion

Despite advancements in cancer therapeutics, current chemotherapeutic agents are still undesirable due to cytotoxicity and low selective toxicity [27]. Therefore, novel approaches, including combination therapies for targeting cancerous cells with minimal side effects, are required. 

Previous studies have reported that Nar or Que exert promising chemotherapeutic effects in MCF-7 BC cells by reducing cell viability and inducing apoptosis [28,29,30,31]. However, high concentrations were required to achieve a minimum inhibitory response. Moreover, both Que and Nar have poor water solubility, which could impact their bioavailability for cellular uptake and further reduce efficacy [32,33]. In previous studies, a high concentration of 780–880 µM (212–239 µg/mL) of Nar was needed to reduce cell viability by 50% (IC_50_) in MCF-7 BC cells [31]. This study used DMSO to dissolve Que and Nar to improve cellular uptake and penetration. An IC_50_ of 468 µg/mL and 91.1 µg/mL were determined for Nar and Que, respectively. However, our results showed that CoQN at a 60:40 ratio produced a significant synergistic effect on MCF-7 cell viability at a lower concentration (44.3 µg/mL) after a 24-h incubation period compared to 40:60, 50:50 ratios and either flavonoid alone, respectively Figure 2A,B. This indicates the synergistic potency of the combination of the two flavonoids.

Chemotherapeutic cytotoxicity by increased apoptosis is an effective method in cancer treatment leading to the reduction of cell proliferation [19]. Caspases are cysteine proteases that play a vital role in apoptosis execution through intrinsic and extrinsic pathways. Caspase 3/7 are known as executioner caspases, whereas caspases 9 and 8 are responsible for caspase induction by the intrinsic and extrinsic apoptotic pathways [34]. Previous studies have reported apoptosis induction by caspase activation following treatment with Nar or Que in different cancers, including BC [11,35,36,37,38,39]. To evaluate the synergistic effects of CoQN on apoptosis induction in MCF-7, the level of caspase 3/7 activity was compared to either Que or Nar alone. Our findings demonstrated an increase in caspase 3/7 activity in MCF-7 after a 24-h treatment period with Nar and Que alone compared to untreated cells (Figure 3). However, CoQN produced significantly higher caspase 3/7 activation compared to either Que or Nar alone at the same concentration (44.3 µg/mL), leading to more profound apoptosis induction and reduced cell proliferation (Figure 3).

Furthermore, the anti-apoptotic Bcl-2 protein activity was investigated by measuring its mRNA expression in the different treatment groups. The downregulation of Bcl-2 is a valuable mechanism for apoptosis induction via the intrinsic pathway [18,34]. Que and Nar have previously been reported to exert cytotoxic effects by regulating several members of the Bcl-2 family of proteins, including Bax, Bcl-xl and Bcl-2, resulting in intrinsic pathway-mediated apoptosis in neoplastic cells [40,41,42,43,44,45,46,47]. Our study shows that Que and Nar cause a significant decrease in Bcl-2 mRNA expression in MCF-7 cells compared to untreated cells (Figure 4). However, our study also showed that Nar or CoQN produced the significant downregulation of Bcl-2 expression compared to either Que or controls, suggesting a synergistic effect leading to intrinsic pathway-mediated apoptosis in MCF-7 BC cells (Figure 4).

The overproduction of intracellular ROS by endogenous or exogenous sources is one of the major driving forces leading to apoptosis induction [48]. ROS are mainly produced in mitochondria as metabolic by-products in biological systems [48]. Enzymatic antioxidants, such as catalase and superoxide dismutase (SOD), break down and remove free radicals. In contrast, non-enzymatic antioxidants, such as carotenoids and glutathione, convert free radicals into hydrogen peroxide (H_2_O_2_) and then water, in the presence of metal cation cofactors [21]. Malignant cells lack some antioxidants, such as catalase, which is present in non-cancerous cells and cannot, therefore, neutralize free radicals.

Furthermore, flavonoids have been shown to exhibit pro-oxidant effects in cancer cells [49]. Some flavonoids can complex with metal cations, leading to the increased production of superoxide ions through Fenton reactions, which induces apoptosis and halts cell proliferation [23]. Quercetin can also undergo auto-oxidation resulting in the formation of the radical, Quercetin-O•, which further leads to free radical accumulation [50] or can diminish glutathione stores and inhibit the enzymatic antioxidant, thioredoxin reductase [51,52]. Naringenin has also displayed pro-oxidant effects in malignant cells by inhibiting glutathione peroxidase and the ROS-dependent activation of ERK1/2 [53,54]. The overall increase in ROS production induces lipid peroxidation and MDA formation, which damages cellular components, such as membrane lipids, and mitochondrial DNA, leading to apoptosis and reduced cell proliferation [55].

The resultant oxidative stress-induced apoptosis has been reported in several neoplastic cells, including BC, hepatocellular carcinoma, cervical cancer and skin cancer [12,56,57,58,59]. In our study, neither Que nor Nar at a lower concentration (44.3 µg/mL) showed a significant increase in MDA production, indicating no significant increase in lipid peroxidation and ROS production. However, CoQN administration at the same concentration showed a significant increase in lipid peroxidation, signifying a synergistic effect resulting in increased ROS production (Figure 5). This agrees with a recent study which reported that high ROS production with Que leads to synergistic chemotherapeutic effects with tamoxifen in MCF-7 cells [56]. Nar and curcumin co-delivery in nanoparticles in MCF-7 cells also increased ROS production and apoptosis induction [60]. Furthermore, increased ROS production and lipid peroxidation has previously been linked to the downregulation of Bcl-2 expression associated with subsequent caspase 3/7 activation leading to apoptosis [61,62,63]. This can be connected to the observed significant reduction in Bcl-2 expression and increased caspase 3/7 activity observed with CoQN treatment (Figure 3 and Figure 4). 

Another marker of potential apoptosis induction is MMP reduction, indicating mitochondrial depolarization [64]. Excess intracellular ROS production is one of the major factors leading to mitochondrial dysfunction and associated decrease in MMP [65]. Antineoplastic effects of flavonoids have previously been reported [58,59,66,67,68]. Therefore, MMP expression was assessed in MCF-7 cells following Que, Nar or CoQN exposure. Results showed that CoQN caused a slightly significant decrease in MMP compared to either Que or Nar, respectively (Figure 6). The synergistic activity of Que with curcumin has previously been reported in a study by Zhang et al. in gastric cell lines, demonstrating a reduction in MMP with subsequent apoptosis induction and reduced cell proliferation [69]. Similarly, Nar and curcumin caused oxidative stress-induced mitochondrial depolarization resulting in apoptosis induction in MCF-7 cells [60]. Therefore, we propose that the observed reduction in MMP following CoQN exposure results from the synergistic pro-oxidant effects of Que and Nar in MCF-7 cells indicated by the significant increase in ROS production. We used 60:40 ratio for Que:Nar, which closely mirrors their relative native concentration ratios in citrus fruits [70] and the enhanced antiproliferative effects could be a manifestation of their biological effects in the native fruit. This approach would give an opportunity for formulation of these flavonoids or their synthetic analogs into effective biopharmaceuticals.

Therefore, it can be concluded that CoQN shows promising synergistic antiproliferative effects in MCF-7 cells compared to either Nar or Que, respectively; by increasing lipid peroxidation, inducing mitochondrial depolarization, anti-apoptotic Bcl-2 suppression and concomitant activation of caspase 3/7. Flavonoids are known to be non-toxic to non-malignant cells [11,71,72]; hence more studies are suggested to confirm their antineoplastic role in this regard.

## 5. Conclusions

The co-administration of the flavonoids Que and Nar shows promising synergistic antiproliferative effects in MCF-7 BC cells compared to either alone by significantly increasing oxidative stress, reducing MMP and inducing apoptosis, leading to cytotoxicity and a reduction in cell viability. Our results here support the notion that the increased consumption of fruits and vegetables rich in these flavonoids is associated with a reduced incidence of many types of cancers. We posit that these flavonoid combinations further be developed as chemotherapeutic agents against breast cancer. Theses flavonoids or their synthetic analogs could further be investigated and delivered in vivo via appropriate vehicles in nanoparticles delivery system as biopharmaceuticals.

## Figures and Tables

**Figure 1 nutrients-14-03437-f001:**
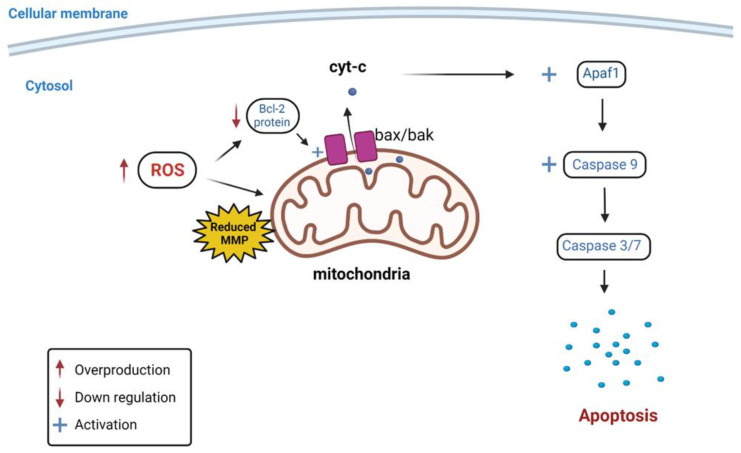
Schematic illustration of the effect of increased ROS production on Bcl-2 protein, MMP, caspase activity and intrinsic apoptotic pathway activation. ROS = Reactive oxygen species; MMP = Mitochondrial membrane potential; cyt-c = Cytochrome c; Apaf1 = Apoptotic peptidase activating factor 1; 
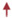
 = Overproduction; 
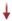
 = Down regulation; 
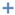
 = Activation.

**Figure 2 nutrients-14-03437-f002:**
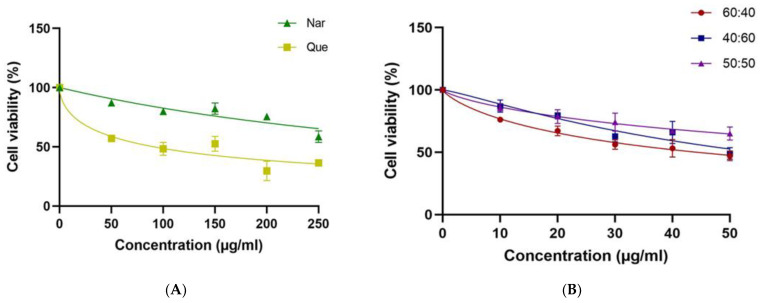
Dose-dependent MCF-7 cell viability as determined MTT assay after the cells were exposed to (**A**): different concentrations of Nar or Que, respectively, and (**B**): different combination ratios of Nar and Que, respectively.

**Figure 3 nutrients-14-03437-f003:**
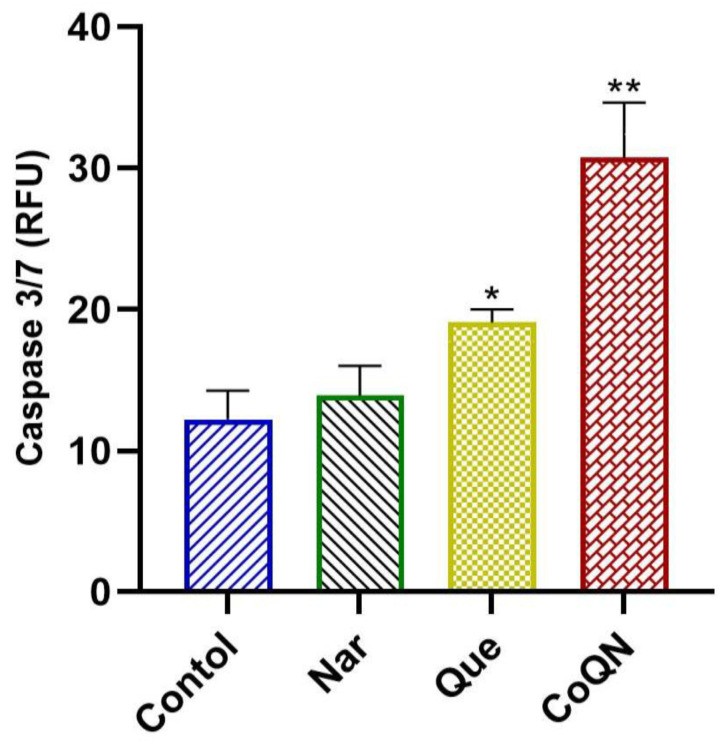
Caspase 3/7 fluorometric assay after MCF-7 BC cells were exposed to 44.31 µg/mL of Nar, Que or both, respectively. * *p* < 0.05 relative to control. ** *p* < 0.01 relative to control. RFU = Relative fluorescence units; Ex/Em = Excitation/Emission; BC = Breast cancer; Nar = Naringenin; Que = Quercetin. CoQN = Co-administration of quercetin and naringenin.

**Figure 4 nutrients-14-03437-f004:**
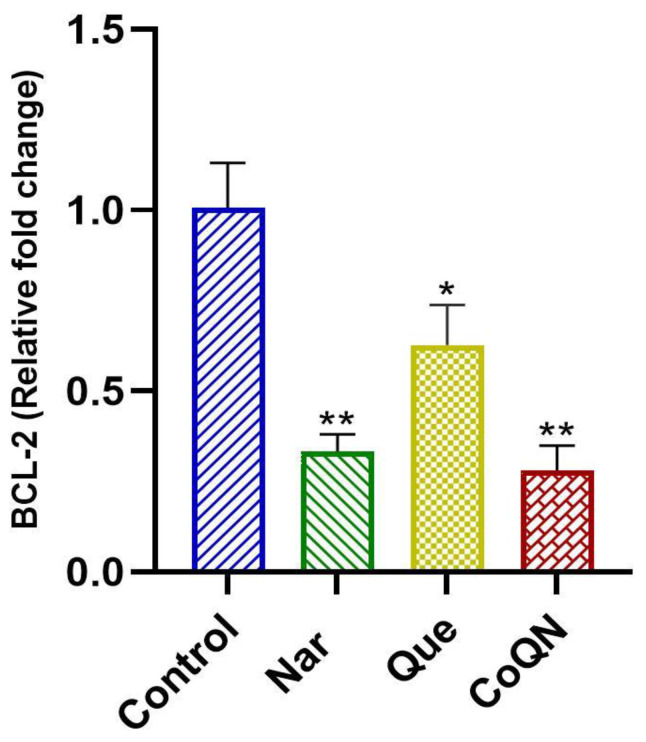
BCl-2 gene expression as determined by qRT-PCR after MCF-7 BC cells were exposed to 44.31 µg/mL of Nar, Que or CoQN, respectively. * *p* < 0.05 relative to control. ** *p* < 0.01 relative to control. BC = Breast cancer; Nar = Naringenin; Que = Quercetin. CoQN = Co-administration of quercetin and naringenin.

**Figure 5 nutrients-14-03437-f005:**
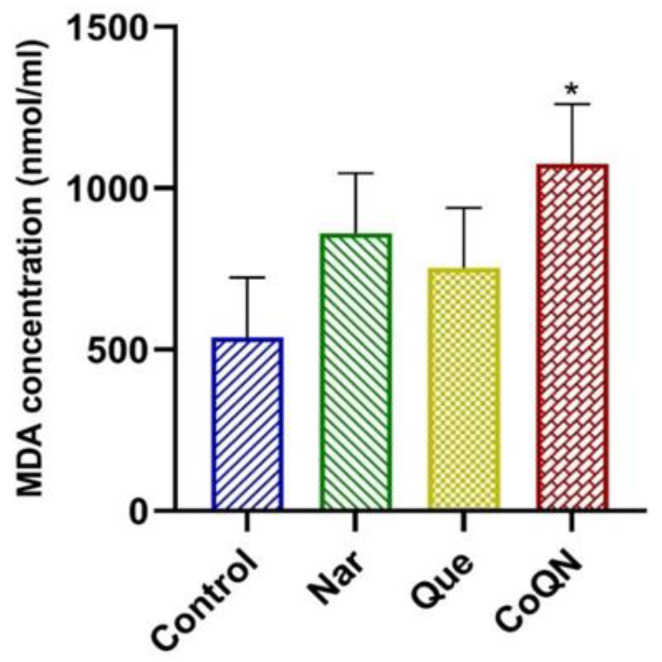
Lipid peroxidation measured by MDA concentrations after MCF-7 BC cells were exposed to 44.31 µg/mL of Nar, Que or CoQN, respectively. * *p* < 0.05 relative to control. MDA = Malondialdehyde; BC = Breast cancer; Nar = Naringenin; Que = Quercetin. CoQN = Co-administration of quercetin and naringenin.

**Figure 6 nutrients-14-03437-f006:**
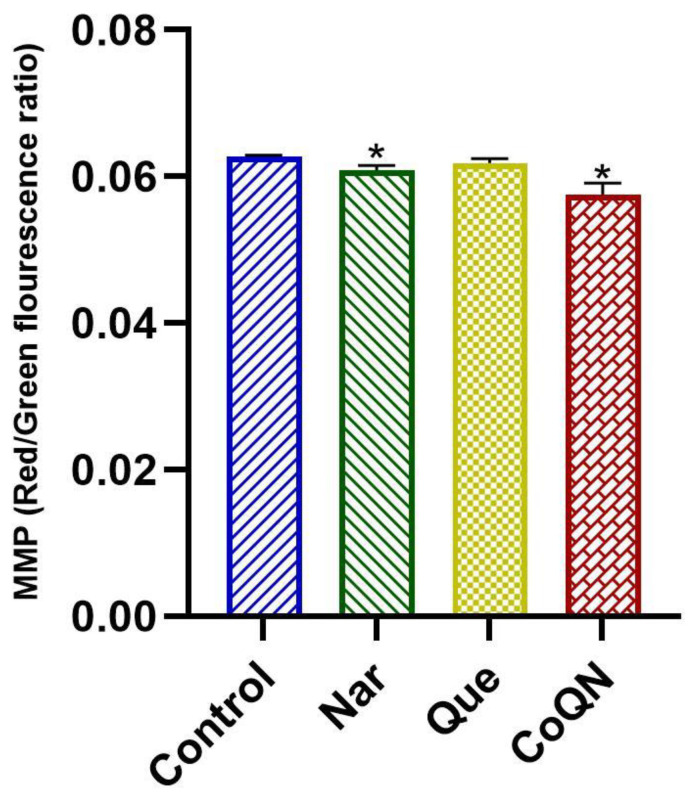
MMP measured by red/green fluorescence ratio after MCF-7 BC cells were exposed to 44.31 µg/mL of Nar, Que or CoQN, respectively. * *p* < 0.05 relative to control. MMP = Mitochondrial membrane potential; BC = Breast cancer; Nar = Naringenin; Que = Quercetin. CoQN = Co-administration of quercetin and naringenin.

## Data Availability

Not applicable.

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
