# Peer review of "Synergism Potentiates Oxidative Antiproliferative Effects of Naringenin and Quercetin in MCF-7 Breast Cancer Cells"

_nutrients, 2022, doi:10.3390/nu14163437_

Round 1
Reviewer 1 Report
This work is excessively extensive, and it lacks a significant novelty effect. It discusses the known facts, and a synergistic action of both factors is expected if comparable concentrations of the individual factors are used and of both together in the Figs 2. The fact that the observed result applies to a single cell line does not mean that consumption of vegetables and fruits has the same effect. These data would suffice for much shorter work. There is a lot of repetition, especially in the discussion where there is information from the introduction and there is a re-discussion of the results. The effect on mitochondrial depolarization is on the verge of method error. Also, no leakage of LDH means that the observed apoptosis, denoted as caspase activation, is at an early stage. The determination of MTT does not define the activity of the mitochondrial respiratory chain and the ROS listed are not all free radicals. Abbreviations are also introduced without explaining their meaning, eg Bax and Bak.
Nothing is visible in the photos from the Trypan test, and it is not appropriate test for cell proliferation. LDH is not only used to mark necrosis. It is also now believed that necrosis is programmed (Feoktistova M, Leverkus M. Programmed necrosis and necroptosis signaling. FEBS J. 2015 Jan; 282 (1): 19-31). There are more such mistakes. The work is not suitable for publication in this form. It should be significantly shortened and redrafted with amendments.
Author Response
Some of the experimental work has been truncated. Whereas the synergistic effects of bioactive flavonoids are known phenomena, no previous studies have been conducted on the synergistic effects of naringenin and quercetin in halting MCF-7 cell proliferation and the mechanisms involved. Herein lies the novelty of this study. We are aware that the study was conducted in a single cell line, and surely that does not mean that consumption of fruits and vegetables may lead to the same effects in vivo, but it may suggest so. We have said that in our discussion, which has also been revamped to minimize statements that appear repetitive.
Mitochondrial depolarization may be on the verge of method error due to the small statistical differences, or it may not be. It is unfathomable that method error in this particular experiment which, like others, was done in triplicates, skewed our results. No, leakage of LDH defined by caspase measurement may suggest that apoptosis is at an early stage, but the cells were incubated for 24 hrs, beyond which their physiological survival is curtailed, so how early is early in this particular case? Ture, there are other mechanisms by which cell necrosis occurs, but LDH, which we chose in this case, is one of them. True MTT does not determine mitochondrial respiratory chain but cell viability. The work has been revamped to comply with your suggestions; thank you:
- the discussion has been shortened
- the trypan blue and LDH assays have been removed
- ROS has been redefined to clarify the free radicals' roles
- Implications of MTT assay have been amended as suggested
Reviewer 2 Report
The paper “Synergism potentiates oxidative antiproliferative effects of 2 naringenin and quercetin in MCF-7 breast cancer cells 3” by Mahasin Abdel Rhman, Nikita Devnarain, Rene Myburg and Peter MO Owira described the synergistic anti-cancer effects of Quercitin and and Naringen on MCF-7 cells. Since the topic is very current and interesting and the experimental design is good, some important concerns are about the MM and the presentation of results.
Section MM
Line 119-128: Trypan Blue Exclusion Assay: the method described by the authors is only qualitative and poorly informative.
The authors can use this test in a quantitative way by detaching the cells and counting them with a Burker chamber, in this way this assay would have a real meaning… otherwise it makes no sense to report it, also because the image has a very poor quality.
Line 104: being Nar and Que purchesed by Sigma company, please specify the code number, Cas number, lot identification.
Line 150: if the The expression level was determined using the 2 −ΔΔCt method is correct to add the bibliography:
Livak KJ, Schmittgen TD. Analysis of relative gene expression data using real-time quantitative PCR and the 2(-Delta Delta C(T)) method. Methods. (2001) 25:402–8. doi: 10.1006/meth.2001.1262.
Furthermore, I suggest to the authors never use a single house-keeping gene, but, as all international guides suggest. please use at least 3 gene references.
Section: Statistical analysis
How many times did you replicate experiments? How many technical replicates in any single experiment? These data lack for all the experiments…information must be improved.
Section Results:
Figure 2: I don’t understand the meaning of section C: chemical structure of Nar and Que is not a result…is an information report by Sigma Company, please remove it.
Figure 3: is not informative please read the comment relative to MM, provide quantitive graph by cell counting.
Figure 4: Which is the parameter studied? The title of ordinate axis is wrong: you have measured caspases..so please change RFU with: Caspase 3/7 (RFU).
Figure 5: Which is the parameter studied? The title of ordinate axis is wrong: you have measured Bcl2 so please change relative fold change with: BCL-2 (relative fold change). Please remove melting curve and carefully read my comment below!
Line 229
the phrase: “Melting curves (Fig 5B) confirmed that the RNAs were not degraded during isolation” is wrong, melting curve is important to check the correctness of the molecular weight of the expected product…so please remove it, and remove the graph because the information is obvious ... if the expected product had a different molecular weight than the expected one it would mean that you are wrong pcr and you have not amplified the right gene, so it would not make sense to quantify a wrong gene ... So I have some doubts about the ability of the authors to master PCR teqnique…please study again the qPCR tecnique!
Figure 7: Which is the parameter studied? The unit measure of this graph is not red/green…but MMP (red/green…), plese change it. I have many doubts about the biological significance of statistical reduction…I suggest to represent the original data and data not normalized on the control…
Figure 7: Which is the parameter studied? The title of ordinate axis is wrong: please change Red/green…with MMP (red green fluorescence ratio…)
Author Response
Thank you for the opening compliments:
- Trypan blue assay has been deleted from the manuscript as per your suggestion
- Code no and Cas numbers have been included in the manuscript as directed.
- Reference for the method has been added.
- It is not possible at this stage to redo the experiments to include more than one housekeeping gene as a reference. We shall keep that in mind for future experiments.
- The experiments were performed in triplicates and repeated 3 times. This information is captured in statistical analysis
- The chemical structure of quercetin and naringenin in Fig 2 have been removed as directed.
- Fig 3 showing the trypan blue assay has been removed to avoid ambiguity.
- The y-axis in Fig 4 (now Fig 3) has been relabelled Caspase 3/7(RFU) as suggested.
- Y-axis in Fig 5 (now Fig 4) has been relabelled BCl-2 (Relative Fold Change) as suggested. The melting curves have been removed.
- Y-axis in Fig 7 (now Fig 6) has been relabelled MMP (Red/Green Fluorescent ratio) as directed. Data has been modified to its original form rather than as a percentage of control to confirm statistical significance.
Round 2
Reviewer 2 Report
The authors have satisfied all my requests